# Implications of Mediated Market Access—Exploring the Nature of Vertical Relationships within the Croatian Wine Industry

**Jana Katunar \*, Marija Kaštelan Mrak**  **and Vinko Zaninović** 

Faculty of Economics and Business, University of Rijeka, 51000 Rijeka, Croatia; marija.kastelan.mrak@efri.hr (M.K.M.); vinko.zaninovic@efri.hr (V.Z.)
\* Correspondence: jana.katunar@efri.hr; Tel.: +385-51-355-165

**Abstract:** The aim of this research was to provide a better understanding of factors influencing the performance of (small) agricultural producers. Considering the importance of agricultural producers, not just for maintaining a steady supply of products but also for the preservation of the population (and cultural tradition) of rural areas, the development of sustainable agricultural business is a matter of public interest. This paper considers wine producers and their market channels, i.e., the factors influencing the relationship between wine producers and market intermediaries, by applying ideas taken from agency theory. We developed a conceptual model with our defined measure of agency costs as the mediator variable between multiple regressors and the firms' financial and non-financial performance as regressions. We used the approach of Baron and Kenny to investigate whether agency costs act as a mediator variable. The data needed to test the proposed conceptual model were collected through questionnaires and contextual interviews with the Croatian wine producers (n = 124). We found that more self-reliance in the distribution process, supported by factors related to the bargaining strength, had a positive influence on wine producers' performance. The results also support the assumption that agency costs act as a full mediator variable between a producer's attributes and its performance.

**Keywords:** wine industry; distribution channels; agency theory; incomplete contract theory; sustainable agriculture

## 1. Introduction

The organization of agricultural production and the structure and nature of the value chain, from production to markets, is becoming a topic of interest for producers, researchers, policy makers, and market regulators. Over the past few years, research efforts have been substantially building up, but issues concerning the bargaining positions and economic welfare of small agricultural producers are still under-researched [1]. Research is often run as empirical studies focusing on specific industries in specific countries. Researchers target patterns in the contributions to and the distribution of value added [2], factors related to market power abuse [3,4], and factors affecting the bargaining power of small scale producers [5–7]. Our study adds one more industry- and country-specific contribution to this area because, as far as we are aware, there have been no attempts to conduct research on the factors that influence the bargaining position of Croatian agricultural producers.

Additionally, we believe our research contributes to this research area by combining a factor approach to bargaining power with agency theory using firm level data acquired through primary research. While agency theory mostly interprets the principal-agent relationship through the business owner-manager relationship, more recent research has also looked at inter firm relationships, especially when it comes to the firms in dependent positions or along the distribution chain. Moreover, agency costs in the distribution chain have been mostly considered from the perspective of the buyer, but in this paper, they are considered from the perspective of the producer as the weaker partner in the Croatian wine industry.

The aim of this paper was to explore some of the factors that shape the relationship between the wine producer and wine distributor(s) and to verify whether agency costs can be detected as a factor affecting the performance of wine producers. Our first objective was to measure how attributes describing a wine producer, such as the size of the wine producer (measured by the number of wine labels and production volumes), distribution channels, reputation, social capital, and access to in-house professional advice (specialist economist employed), influence agency costs. Our next objective was to estimate the impact of agency costs on wine producer's performance. The overall statistical framework that we employ is that of Baron and Kenny [8].

The paper is organized as follows. After a brief introduction to our research, we provide information on the trends and challenges faced by Croatian wine producers needed for the understanding of the context behind their choice of distribution strategy. The second part of the paper provides a review of the previous theoretical and empirical literature. The third part of the paper describes and defines the research methodology, and the fourth part presents the research results. The paper ends with discussion and conclusion.

The Croatian wine sector has significantly changed in the last 20 years. Wine production in Croatia was stable during the first decade of the 21st century, after which it began to steadily decline from 1,433,000 hl in 2010 to 733,000 hl in 2018 [9], with imports doubling in size over the last 10 years [10]. Croatia's expectations of an increase in wine exports, due to its accession to the EU in 2013, did not happen regardless of the increase in exports in the decade before Croatia joined the EU. Specifically, its exports stagnated and even declined while imports doubled [11]. Although there are numerous reasons for this, one of them is surely that the wine industry's growth have reached its limits due to the barriers faced by wine producers, such as land limitations, disorganized land ownership books, and limitations caused by planting on plots of lower quality.

By joining the European Union, Croatia became a part of the Common Agricultural Policy (CAP), which is aimed at increasing the efficiency of the agricultural sector. Therefore, both taxation policies and CAP contribute to strengthening the producers' position and provide a foundation for sustainable agricultural business [12]. Through the National Wine Sector Assistance Program, the Croatian wine producers are offered opportunities for further developing their business, increasing the quality of their wines, and increasing the visibility of Croatian wines on the EU market. Furthermore, tourism also has a great impact on the wine sector. Specifically, wine producers from the tourist area, who can arrange to place a part of the production on the market through their own direct-contact market channels and therefore avoid intermediaries, achieve better selling prices and better business results [13]. As their product is often seen as a luxury good, some have been capable of building distinctive brands and establishing reputations for offering high quality wines, at least on the national market.

Croatian wine producers face a number of limitations and challenges in the EU wine market. Considering that they are typically small family businesses (over 99% of wine producers cultivate an area of less than 5 ha) operating with limited resources (limited financial capacity and work force) and offering a narrow range of wine varieties, scaling the business is not a likely possibility for most producers. One of the greatest challenges faced by producers is placing their products on the market. In theory, wine producers have alternative possibilities to organize the distribution of their products. They can organize the distribution on their own, providing they have managed to build a strong reputation, operate a higher volume, possess the necessary financial strength, and be determined to have high control over the distribution process [14]. However, self-reliant distribution strategies imply high costs, such as extensive investment in capacity building, human resources, administration, increased warehousing costs, costs of running and controlling distribution activities, and high uncertainty and unknown risk. Consequently, most wine producers rely on the services of specialized distributors. The relationship between wine producers and distributors is governed by standard contracts. In this paper, we address some of the implications of these contractual arrangements.

In doing so, we proceed from findings and observations provided by other researchers of agricultural producers' access to markets and the need for a better understanding of factors that influence the bargaining positions of small agricultural producers in dealing with large distributors. A recent review of research dealing with links between small farmers and contract and non-contract actors along the market channel in developing countries, spanning 202 studies published since 2000, revealed a constant increase in the number of published papers on the topic (40% published since 2015 and 80% published since 2010); however, there were a very limited number of studies addressing bargaining power among these [1]. In the EU, more attention is provided to actors' relationships along the agricultural and food value chains. Lawyers often deal with the issue under the umbrella of (unfair) trading practices [5], while economic analysis is used to address imperfect competition and factors affecting bargaining positions [6] and value distribution, as in the work Velasquez and Buffaria [7], where the focus was on imperfect price transmission along the food chain.

## 2. Theoretical Background

Agency theory, as proposed by Jensen and Meckling [15], assumes that the principal and the agent act in their own interest. In making decisions that concern the principal, the agent will not necessarily choose the course of action that is in the best interest of the principal; rather, he will pursue decisions that favor his own (the agent's) interest [15]. Therefore, in contracting with the agent, the principal's objective is to ensure that the agent works in the principal's interest at minimum cost. To minimize deviations from its own interest, the principal will attempt to monitor the agent's behavior and use a contract to regulate agent's behavior and establish a compensation scheme to bring the agent's behavior, i.e., the agent's decisions, more in line with the principal's interest.

Theory does not consider perfect contracts to be a situation possible in the real world. What is expected is contract "incompleteness" under conditions of bounded rationality. The problem generally arises from the fact that "unforeseen contingencies" may arise and that contract parties may be aware of these possibilities, but these contingencies are not specified in the contract for reasons of efficiency. "Most real contracts are vague or silent on a number of significant matters" [16] (p. 83). Contract incompleteness potentially leads to ex-post costs that are attributed to incomplete information [17].

Contract incompleteness is a standard aspect of agency situations, as the agency stresses the vertical aspect of the relationship among contracting parties, where the principal actually transfers some of his/her control rights to the agent. This enforces information incompleteness but also asymmetry, which is practically a requirement for the agency problem to occur. It is assumed that one contracting party (in this case the agent) possesses certain information that the other party does not, which leads to opportunistic behavior and moral hazard as likely consequences. To alleviate the agency problem, the principal must reduce/eliminate the conflict of interest by creating an incentive system that leads to an alignment of interests between the principal and the agent. However, bounded rationality and costs of contract design [15] prevent principal-agent contracts from ever becoming perfect.

Jensen [18] argued that it is impossible to ensure that agents will make decisions that are in the best interest of the principal at zero cost. As explained by Maskin and Tirol [16] (p. 84), providing a more comprehensive and case-specific formulation of parties' expectations and contingencies would not be economically viable because "costs prevent agents from describing physical contingencies ex-ante". Agency theory mostly focuses on the costs that arise as a result of the abovementioned solutions, the types of agency costs and their relationship [15], and issues in dealing with the distribution of agency costs.

According to Ramakrishnan and Thakor [19] and Gogineni et al. [20], agency costs (monitoring cost, bonding costs, and residual loss) are exclusively borne by the principal, while some authors have argued that part of the agency costs is borne by the agent itself. Since the agent's income also depends on the business outcome, the agent may be punished

or rewarded for the outcomes (some of which may be beyond his control), so the interests of the principal and the agent overlap [21]. Perow [22] pointed out that agency problems can also occur on the side of the principal whose opportunistic behavior and manipulation can deceive the agent.

The concept of incomplete contracts is commonly applied in the economic analysis of the bargaining positions of parties involved in long-term relationships [23]. Some authors have investigated the producer-buyer/buyer-supplier relationship, and other have considered agency costs from the buyer's perspective [24–28]. Information asymmetry is seen in the observed relationship in favor of a supplier/producer who has product information that the customer/buyer does not have, leading to the possibility of opportunistic behavior. Due to the buyers' need to monitor the principal's behavior, agency problems and thus agency costs occur. The relationship of cooperation between the producer and the distributor, which is focus of this paper, was researched by Lassar and Kerr [29]. In their work, their findings suggest that "strategic choice may provide a useful basis for understanding important aspects of manufacturer-distributor relationship" (p. 628).

The effect of agency costs on firm performance has been present in the academic literature since Michael Jensen's paper "Agency Costs of Free Cash Flow, Corporate Finance, and Takeovers" [18]. More recently, the topic has been updated, and while Wang [30] concluded that the relationship between agency costs and firm performance is inconsistent, Jabbary et al. [31] successfully demonstrated a relationship between agency costs and firm performance on a sample of 73 companies listed on the Tehran Stock Exchange.

## 3. Model Development

This research was designed as a single industry research, which is a common approach in the field [3,4,8].

Within this paper, the authors use agency theory to investigate the relationship between agricultural producers and stronger actors in the distribution chain. The presence of bounded rationality and the opportunism was assumed [32], as was a situation of information and power asymmetry that may result in business decisions that are not necessarily in the best interest of the wine producer. A point made by Cordero Salas inspired us to explore the mediating role of agency costs and their impact on the financial outcomes of wine producers [33]. An illustration of this research model is presented in Figure 1, which is followed by a more detailed explanation of the reasons for setting our research hypotheses.

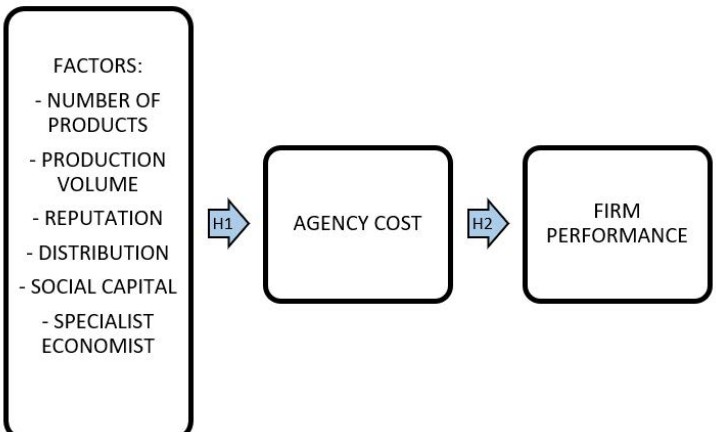

**Figure 1.** Conceptual model.

The case of wine producers presents a typical example of contract incompleteness under asymmetric positions of power. Information asymmetry is expected to benefit the distributor as the "stronger" party. The reason is that the distributor possesses a more powerful bargaining position, considering that he/she has superior market insights, a broader range of professional personnel, more knowledge and experience in negotiation

techniques, knowledge of business processes, and (most importantly) the "entry ticket" to retail chains and restaurants. The negotiation process between the mentioned actors affects the distribution of profits and the price paid by the end customer.

Contracts signed between wine producers and distributors are typically standard and simple, leaving out many important contingencies that eventually reflect on the financial performance of wine producers. Such contracts are not "relation-specific"; from the distributor's viewpoint, logistics, warehousing, and sales comprise a standardized process. The standardization of contracts serves to lower the administration costs borne by distributors and to give them more maneuver space when providing services to wine-producers. The fact that the form of the contract is designed to serve the interests of the distributor reinforces the assumption of power asymmetry in the relationship between the wine-producers and the distributor.

In contrast to the distributor's "standardized" approach, the perspective of the wine producer may be quite different. On the market, wine is not a homogeneous good because there are numerous opportunities for a strategic differentiation among wine producers [34]. Moreover, wine is perceived as an experience product rather than a commodity [35]. Wine prices on end-markets are rather elastic, depending on multiple factors such as wine origin, category, producers' reputation, point of sale (location), occasion, presentation, and currently dominating tastes. Therefore, the possibility to manage contract conditions could grant producers access to better business performance.

By signing the contract with a distributor, the control over strategically important issues, such as the choice of the target market (in geographic or sociodemographic terms), the selling price, the extent and form of advertisement, and the type/attractiveness of sales establishment(s), will be implicitly transferred to the distributor. This observation justifies our approach in treating the relationship between wine-producers and distributors as being an agency relationship. Thus, seen from the perspective of the agricultural producer, the distributor assumes the role of an agent in a principal-agent relationship and the deployment of price and non-price market strategies of the agricultural producer will be mediated by the decision taken by the distributor. Accordingly, since the small agricultural producer engages an "agent" to act as an intermediary in accessing final consumption markets, our expectations were that agency costs will act as a mediating factor between the wine producers' attributes and its financial performance.

### 3.1. Variables and Hypotheses

There are two main hypotheses implied by this model:

**Hypothesis 1 (H1).** *Specific factors attributed to producers affect agency costs.*

**Hypothesis 2 (H2).** *Agency costs reflect on the wine producer's performance.*

These hypotheses were established to explain whether factors attributed to bargaining power (in market transactions) can be associated with a higher degree of agency costs. With our first hypothesis (H1), we examined the influence of factors related to the wine producer's bargaining position on agency costs. The independent variables whose impact were as follows: number of products, production volume, reputation, number of distribution channels, social capital, and employment of a specialized economist. The choice of independent variables was intended, at least when starting this research, as an attempt to establish and measure factors that could be traced to some basic forms of agency costs. As a result, the act of hiring a specialized economist by the wine-producer may be interpreted, at least partly, as a cost undertaken in order to have better control or rather to create some awareness of how well the distributor is tending to the wine-producer's interests. Therefore, hiring a specialist could be considered a monitoring cost. However, as our research (the interviews) proceeded, it became more apparent that there was no possibility of classifying agency costs due to the small scale of wine producers and the standard form of contract agreements. Such reasoning led us to assume that the interests served by the distributor would be primarily his own. The "level of service" that favors the interests

of wine-producers eventually depends on the particular attributes of the wine-producer himself/herself. The attributes representing factors to be tested are explained in more detail bellow. Consequently, all agency costs—monitoring, bonding, and residual costs—were assumed to be included in the distributor's margin.

Therefore, in order to measure the impact of agency costs on performance in the regression model, the difference in dealer and retail price increased by monitoring costs was treated as a reflection of agency costs, which leads to our second hypothesis (H2).

The main hypotheses were followed by sub-hypotheses:

**Hypothesis 1.1 (H1.1).** *The number of products (labels) influences agency costs.*

**Hypothesis 1.2 (H1.2).** *The volume of production influences agency costs.*

The first two variables—the number of products (measured though the number of labels offered by the wine producer; H1.1.) and production volume (expressed as the annual production in liters; H1.2.), taken from the perspective of the distributor—were expected to produce the equivalent of scale economies; it can be expected that the distributor will have higher earnings per unit as a result of efforts made in administering a relationship with a specific wine producer while the volume and variety of products increases. Therefore, the expectation is that these two factors provide a negative impact on wine-producers' agency costs, which transforms into a positive influence on their performance. It should be noted that the previous research we consulted mostly dealt with bargaining positions and not agency costs, although Ang et al. [36] controlled for firm sales, which, given the specifics of wine industry, is comparable with the volume of production variable that we chose. The previous studies of the impact of firm size on firm performance showed inconsistencies. Though Becker-Blease [37] proved a negative relationship between the observed variables, Pervan and Višić [38], Akinlo [39], Dogan [40], and Babalola [41] proved a positive impact of firm size on financial performance.

**Hypothesis 1.3 (H1.3).** *Reputation influences agency costs.*

Reputation, as a factor, is expected to facilitate sales' volume and lower the costs of advertising borne by the distributor. Although reputation is not easy to quantify because it is determined by perception and interpretation of the observer, many scientific studies have addressed the impact of reputation on product price [42,43] and firm financial performance [44–48]. Reputation also plays an essential role in solving the problems arising from information asymmetry [49] and reduces the incentives of opportunistic behavior, thus reducing the costs of monitoring and bonding [50]. Moreover, Meuleman at el. [49] tested the role of reputation in alleviating agency costs.

**Hypothesis 1.4 (H1.4).** *Distribution influence agency costs.*

The variable distribution, used by H1.4. is a factor assumed to have a different effect compared to the previous three factors. Reliance on the distributor was chosen as an indication of the level of the wine producers' dependency. Our supposition was that the possibility to access alternative distribution channels would grant the wine producer more bargaining power, making the distributor more concerned with maintaining the contract relationship. The impact of distribution channels on firm performance was studied by Lassar and Kerr [29], who concluded that the greater the number of distributors in a geographic area, the greater the loss of control for the produce, and the higher the possibility of opportunistic behavior by distributors. On the other hand, Coelho et al. [51] proved a positive relationship between the number of distribution channels and sales performance.

The last two factors, social capital (H1.5.), and internalized professional advice addressed though the question on whether the wine-producer employed an economist (H1.6.), were as indicators of the wine producer's negotiating capacity. As with distribution, these two variables were intended to measure the bargaining power of the wine-producer.

**Hypothesis 1.5 (H1.5).** *Social capital influences agency costs.*

The relationship between social capital and agency theory was examined in the work of Beccera and Gupta [52], who concluded that the lower the social capital (measured by trust), the higher the agency costs. According to Pospech and Spešna [53], social capital, especially informal personal ties, has a positive influence on the economic performance of firms in the agricultural sector. In addition to these, there have been a number of other studies analyzing the relationship between social capital and firm performance [54–56].

**Hypothesis 1.6 (H1.6).** *Employing a specialist (economist) influences agency costs.*

In the reviewed literature, we found that studies of human capital on firm development have been mainly conducted on samples including large companies (due to data availability), while Storey [57] concluded that investing in specialized workforce in small firms is riskier and has a significantly lower expected return.

Our main hypothesis states that agency costs influence the performance of wine producers.

It was assumed that agency costs are borne by the wine producer. For previously mentioned reasons and because there was no available data estimation of their monetary value, we treated agency costs as a single entity. Their expected impact on performance was negative. The sub-hypotheses assumed that financial performance is measured through revenue per employee (H2.1.) and non-financial performance is measured through prizes won (H2.2.).

**Hypothesis 2.1 (H2.1).** *Agency costs influence financial performance (revenue per employee).*

**Hypothesis 2.2 (H2.2).** *Agency costs influence non-financial performance (number of prizes won).*

The literature connected to these hypothesis (primarily with H2.1.) includes the work of Wang [30], who investigated the impact of agency costs on firm performance (that is, on stock return) and that of Cadot [58], who investigated the impact of agency costs on the financial performance of firms, namely revenues.

### 3.2. Materials and Methods

The data gathering for empirical analysis consisted of two phases. The first phase involved a questionnaire. The questionnaire consisted of 45 questions that were divided into four parts. The first part of the questionnaire consisted of five general questions about the company and its production. The second and the third part of the questionnaire consisted of 37 questions to be used for the quantitative analysis (open-answered questions requiring numerical answers and 1–5 Likert scale questions ranging from 1—distinctly disagree to 5—strongly agree). The fourth part of the questionnaire consisted of 3 open-end questions asking respondents to give their opinions concerning the company's business plan, their opinion about the strengths and weaknesses of the Croatian wine industry, and the perspective of the wine industry through the EU funds and joint appearance on the foreign market. The second phase of data gathering consisted of contextual interviews with wine producers based on the completed questionnaires, which were very important for the interpretations of the results. The interviews with 124 wine producers were conducted during 2019 before the COVID-19 crisis began. The target group consisted of wine producers who sell part of their production through distributors.

### 3.2.1. Data and Sample Description

The 124 wine producers who were interviewed were found to cultivate 6053 hectares of land, which represents more than 31% of the total area under vineyards in Croatia [59] and, according to our research, represent almost 50% of the total number of producers who distribute their products through an intermediary (distributor). Wine producers registered as sole proprietorship, family farm, and private limited company were found to account for 92% of the sample and were almost equally represented; meanwhile, cooperatives accounted for 2.4% of the sample, and corporations (public limited companies) accounted for 5.6%. If we divide Croatia in two main wine regions, Continental and Adriatic, wine

producers from the Continental region comprised 44.4% of the sample and those from the Adriatic region comprised 55.6%.

Table 1 shows all variables used in our research and their definition, while Table 2 shows descriptive statistics of all variables used in the model.

**Table 1.** Variable operationalization.

| Variable | Acronym | Definition |
|---|---|---|
| Number of Products | nrp | Size of a wine producer is expressed through the number of wine labels (economies of scope) |
| Production Volume | vol | Size of a wine producer is expressed through the production volume in liters (economies of scale) |
| Reputation | rep | Reputation ranges from 1 to 5 on the Likert scale and implies the estimated recognizability of the wine producer on the Croatian wine market |
| Distribution | distr | % of sale through the intermediary |
| Social Capital | sc | Social capital ranges from 1 to 5 on the Likert scale and implies an estimated strength of the impact of the business owner or person in charge of negotiation on the inclusion of wine in the distributor's offer |
| Specialist Economist | se | Existence of the economic expert, educated for negotiation, marketing, and brand building-(dummy variable) |
| Agency Cost | agc | The difference between the price to the distributor and the average retail price, increased by monitoring cost and expressed as a % of the retail price |
| Revenue per Employee | rpe | Revenues per employee, in kunas |
| Prizes | prizes | Number of prizes won in wine producers competitions |
| Region Dummy | jadran | Regional dummy with value 1 if the wine producers is situated in Jadranska (Adriatic) Hrvatska, 0 if it is in Kontinentalna (Continental) Hrvaska |

Source: Authors' work.

**Table 2.** Descriptive statistics.

| Variable | Acronym | N | Mean | Std. Dev. | Skewness | Kurtosis |
|---|---|---|---|---|---|---|
| Number of Products | nrp | 124 | 9.725806 | 6.364713 | 2.651662 | 12.74236 |
| Production Volume | vol | 124 | 316,209.7 | 1,041,232 | 6.351775 | 46.99786 |
| Reputation | rep | 124 | 3.537634 | 0.955590 | −0.094476 | 2.207173 |
| Distribution | distr | 124 | 79.71774 | 19.39557 | −1.286389 | 4.657992 |
| Social Capital | sc | 124 | 3.975806 | 0.932378 | −0.677155 | 2.907829 |
| Specialist Economist | se | 124 | 0.387097 | 0.489062 | 0.463586 | 1.214912 |
| Agency Cost | agc | 124 | 51.49194 | 7.300458 | 0.513243 | 2.815672 |
| Revenue per Employee | rpe | 124 | 544,493 | 324,841.2 | 3.338189 | 22.23831 |
| Prizes | prizes | 124 | 3.320161 | 2.57224 | 1.171408 | 3.971669 |
| Region Dummy | jadran | 124 | 0.5564516 | 0.4988184 | −0.2272596 | 1.051647 |

Source: Authors' calculation.

### 3.2.2. The Empirical Models

According to the descriptive statistics, the dependent variable of agency cost (agc) was normally distributed. Moreover, we tested revenues per employee (rpe) for normality, and the tests (Jarque-Bera and Royston) showed that the variable was not normally distributed. We then checked for the normality of the log-transformed variable, and the null hypothesis of normal distribution could not be rejected (*p*-values of 0.1562 and 0.1490 for previously mentioned normality tests). Therefore, we performed another robustness check of our results, as shown in Table 3, where we used log-transformed rpe. The results (significance of the estimated coefficients) were in line with the original estimation results.

**Table 3.** Results of the four step estimations for the financial performance indicator.

| Variables | (1) rpe | (2) agc | (3) rpe | (4) rpe |
|---|---|---|---|---|
| nrp | 1820 | −0.0223 | | 1581 |
| | (9381) | (0.101) | | (9925) |
| vol | 0.0261 | $6.76 \times 10^{-7}$ | | 0.0334 |
| | (0.0303) | ($1.09 \times 10^{-6}$) | | (0.0328) |
| rep | 75,649 * | −2.074 *** | | 53,359 |
| | (44,436) | (0.725) | | (41,851) |
| distr | −37,665 | 10.90 *** | | 79,507 |
| | (107,969) | (2.944) | | (119,464) |
| sc | −28,244 | −1.110 | | −40,177 |
| | (44,495) | (0.686) | | (45,999) |
| se | 87,488 | −3.907 *** | | 45,501 |
| | (62,283) | (1.113) | | (59,158) |
| agc | | | −12,345 *** | −10,745 *** |
| | | | (2957) | (3657) |
| Constant | 359,278 *** | 56.11 *** | $1.181 \times 10^{-6}$ *** | 962,210 *** |
| | (135,117) | (3.676) | (167,781) | (274,560) |
| Observations | 124 | 124 | 124 | 124 |
| R-squared | 0.082 | 0.290 | 0.077 | 0.124 |

Robust standard errors in parentheses; *** $p < 0.01$, ** $p < 0.05$, * $p < 0.1$; Source: Authors' estimations.

To empirically test our hypothesis and specifically investigate the validity of our conceptual model (Figure 1), we followed the approach of Baron and Kenny [1]. They proposed a four step approach, with each step representing a regression model that distinguishes between exogenous variables (X), mediation variables (M), and dependent variables (Y). In our case, X is an n *k matrix with k independent variables (number of products (nrp), production volume (quant), number of distribution channels, reputation, a proxy variable for social capital (sc), and the dummy specialized economist (se) with a value of 1 if the wine producer employs educated economist and 0 if not), M is the mediation variable, and Y is a proxy variable for firm performance (revenues per employee (rpe) as a financial indicator and prizes as a non-financial indicator). We developed and estimated four regression models:

1.  Regression of X on Y.
2.  Regression of X on M.
3.  Regression of M on Y.
4.  Regression of both X and M on Y.

The goal of these four steps was to determine the (significance of the) relationships between the variables. If one or more relationships among the first three were not significant, we could conclude that the existence of mediation was not very likely. However, if we found in step 4 that, after controlling for M, the variables in the matrix X were not significant, we could conclude that there is full mediation. We later show that this result was unambiguous in our case, i.e., agency costs were found to comprise a mediator variable between X and firm performance. We estimated all models using the ordinary least squares estimator.

The described models can be written as follows:

$$rpe_i = \beta_0 + \beta_1 nrp_i + \beta_2 quant_i + \beta_3 distr_i + \beta_4 rep_i + \beta_5 sc_i + \beta_6 se_i + u_i \qquad (1)$$

$$agc_i = \beta_0 + \beta_1 nrp_i + \beta_2 quant_i + \beta_3 distr_i + \beta_4 rep_i + \beta_5 sc_i + \beta_6 se_i + u_i \qquad (2)$$

$$rpe_i = \beta_0 + \beta_1 agc_i + u_i \qquad (3)$$

$$rpe_i = \beta_0 + \beta_1 nrp_i + \beta_2 quant_i + \beta_3 distr_i + \beta_4 rep_i + \beta_5 sc_i + \beta_6 se_i + \beta_7 agc_i + u_i \qquad (4)$$

Moreover, we also went through the aforementioned procedure with an included regional dummy (a dummy variable called Adriatic, with a value of 1 if the wine producer

was from the Adriatic region in Croatia and zero otherwise), since the regional effects are especially strong in Croatia and we wanted to test whether there is a difference in bargaining power between wine producers in the Adriatic region in comparison with those in other regions. Finally, we performed a robustness check of our results in the way that we performed a confirmatory factor analysis, thus creating one factor variable out of six independent variables used in Equations (1), (2) and (4) (agency costs are separately considered in Equation (4)). We also estimated Equations (1)–(4), with the non-financial indicator (prizes) as the dependent variable instead of rpe.

If we compare our hypothesis with Equations (1)–(4), we can see that H1 is tested by Equation (2), while H2 (as well as sub-hypotheses H2.1. and H2.2.) is tested by Equations (3) and (4). Sub-hypothesis H1.1.–H1.6. are tested by Equation (1).

## 4. Results

The hypotheses were tested by a regression analysis as described in the previous section. If we contrast the results in Tables 3 and 4 with the hypotheses, we can observe that H1 and H2 were generally confirmed. If we look at the level of sub-hypothesis, we see that H1.3., H1.4., and H1.6. were statistically confirmed, namely that specific firm factors such as reputation, distribution channels, and having a specialist economist were found to influence agency costs (they lower agency costs—see column 2 in Tables 3 and 4). Hypothesis 2.1. and 2.2. were also statistically confirmed—increases in agency costs (negatively) were found to influence both the financial and non-financial performance of the firm (see columns 3 and 4 in Tables 3 and 4).

**Table 4.** Results of the four step estimations for the non-financial performance indicator.

| Variables | (1) Prizes | (2) agc | (3) Prizes | (4) Prizes |
|---|---|---|---|---|
| nrp | 0.146 *** | −0.0223 | | 0.145 *** |
| | (0.0292) | (0.101) | | (0.0285) |
| vol | $6.09 \times 10^{-7}$ *** | $6.76 \times 10^{-}$ | | $6.56 \times 10^{-7}$ *** |
| | ($1.41 \times 10^{-7}$) | ($1.09 \times 10^{-6}$) | | ($1.32 \times 10^{-7}$) |
| rep | 0.685 *** | −2.074 *** | | 0.542 ** |
| | (0.261) | (0.725) | | (0.247) |
| distr | 0.124 | 10.90 *** | | 0.875 |
| | (1.079) | (2.944) | | (1.098) |
| sc | −0.125 | −1.110 | | −0.201 |
| | (0.223) | (0.686) | | (0.219) |
| se | −0.126 | −3.907 *** | | −0.396 |
| | (0.378) | (1.113) | | (0.408) |
| agc | | | −0.0860 *** | −0.0689 ** |
| | | | (0.0310) | (0.0281) |
| Constant | −0.269 | 56.11 *** | 7.752 *** | 3.598 * |
| | (1.074) | (3.676) | (1.614) | (1.959) |
| Observations | 124 | 124 | 124 | 124 |
| R-squared | 0.407 | 0.290 | 0.059 | 0.434 |

Robust standard errors in parentheses; *** $p < 0.01$, ** $p < 0.05$, * $p < 0.1$; Source: Authors' estimations.

The results of our estimations (Table 3) also showed that agency costs comprise a full mediator variable between our selected independent variables in the conceptual model (Figure 1) and firms' performance indicators, such as sales per employee. Moreover, we found that the impact of agency costs on the firm performance indicator was negative, which was consistent with our expectations, as well as theoretical and empirical results. This strengthens the case for preferring direct sales of wine and is consistent with the findings of Newton, Gilinsky, and Jordan [34], who studied the US market and found that DTC generates higher gross margins. This is because when wine producers sell their wine to distributors and wholesalers, they typically sell it at 50% of the final retail price (in our sample, this percentage represented agency costs and equaled 51.5%, as shown in Table 2).

On the other hand, Table 4 shows that in the case of dependent variable being the non-financial indicator, i.e., the number of awards won, agency costs were partial mediators. In that case, the reputation variable comes to the center of the analysis of the results, as we could see clearly that the coefficient of that variable has positive and significant influence on the number of awards won, which was in line with our expectations. Moreover, the significance of that coefficient, as well as that of the number of products (np) and volume produced (vol), was not lost when we included our agency costs proxy (column 4 of Table 4).

The results in Table 5 show that once when we controlled for different wine-producing regions in Croatia, the Adriatic and Continental regions, the significance of reputation variables disappeared. We emphasize that the number of wine producers coming from these two regions was almost the same in our sample. We also noticed that revenues per employee, which can be used a proxy of firm productivity, was higher for wine producers in the Adriatic region.

**Table 5.** Results of the four step estimations with the dummy variable accounting for two wine-producing regions.

| Variables | (1) rpe | (2) agc | (3) rpe | (4) rpe |
|---|---|---|---|---|
| nrp | 3477 | −0.0569 | | 3029 |
| | (8502) | (0.116) | | (9140) |
| vol | 0.0436 | $3.11 \times 10^{-7}$ | | 0.0460 |
| | (0.0300) | $(1.09 \times 10^{-7})$ | | (0.0314) |
| rep | 42,165 | −1.374 ** | | 31,353 |
| | (44,623) | (0.667) | | (43,090) |
| distr | 51,754 | 9.033 *** | | 122,853 |
| | (117,010) | (2.943) | | (125,155) |
| sc | −25,435 | −1.169 * | | −34,639 |
| | (43,700) | (0.626) | | (45,589) |
| se | 50,895 | −3.142 *** | | 26,167 |
| | (59,577) | (1.154) | | (57,249) |
| agc | | | −8588 ** | −7871 ** |
| | | | (3291) | (3900) |
| 1. Adriatic | 179,029 *** | −3.747 *** | 132,005 ** | 149,533 *** |
| | (53,123) | (1.179) | (57,222) | (56,007) |
| Constant | 288,231 ** | 57.60 *** | 913,574 *** | 741,601 *** |
| | (130,575) | (3.720) | (190,293) | (279,154) |
| Observations | 124 | 124 | 124 | 124 |
| R-squared | 0.141 | 0.341 | 0.111 | 0.162 |

Robust standard errors in parentheses; *** $p < 0.01$, ** $p < 0.05$, * $p < 0.1$; Source: Authors' estimations.

Finally, Table 6 presents the results of the robustness check of our main econometric models/procedures. For Table 6, we used factor analysis to obtain one synthetic variable that represented our six main independent variables. It is clear from results that the conclusions stayed the same, i.e., agency costs were the full mediator variable in the case of the financial performance indicator and the results did not confirm mediation in the case of the non-financial performance indicator (results not shown here but are available upon request).

Finally, because of our mediator variable testing procedure, we could also observe the effects of inputs on agency costs. This can be seen in column 2 of Tables 3–5. From these results, we clearly observed that increase in the percentage of sales through intermediary raises agency costs, while reputation and having a specialist economist lower agency costs, which was in line with theoretical expectations.

**Table 6.** Results of the robustness check of the main model.

| Variables | (1) rpe | (2) agc | (3) rpe | (4) rpe |
|---|---|---|---|---|
| srm | 35,273 ** | −1.777 *** | | 14,922 |
| | (14,315) | (0.517) | | (16,401) |
| agc | | | −12,345 *** | −11,450 *** |
| | | | (3880) | (3471) |
| Constant | 544,493 *** | 51.53 *** | $1.81 \times 10^6$ *** | $1.135 \times 10^6$ *** |
| | (28,980) | (0.621) | (201,925) | (194,321) |
| Observations | 124 | 124 | 124 | 124 |
| R-squared | 0.021 | 0.107 | 0.077 | 0.080 |

Robust standard errors in parentheses; *** $p < 0.01$, ** $p < 0.05$, * $p < 0.1$; Source: Authors' estimations

## 5. Discussion

Building on the existing research on the impact of agency costs on firm performance, the authors of this paper examined the hypothesis that a wine producer-distributor relationship that incurs higher agency costs for the producer will lower producer's financial and non-financial performance.

Considering the fact that the Croatian wine industry growth has been slowed down in the past few years led to the additional consideration that the possibility that the development of the sector has been hurt by the development of growingly asymmetric positions of wine producers towards distributors.

In preparing this research, we assumed that the bargaining position of small wine producers, when confronted with the distributors' market power, is somewhat similar to the position of small shareholders in a large corporation (as in the model of outside shareholders with no voting rights; see the work of Jensen and Meckling, 1976). Consequently, we presumed that agency theory could best serve our purpose of establishing whether and to what extent the position of small wine producers is similar to the position of minority shareholders. Several circumstances seem to support such similarities, such as the prospects of many depending on the interest of a single/few effective decision makers and standard contracts not designed to meet the specific circumstances of individual wine producers.

It is interesting to note that our results (column 4, Table 3) showed that an increase in agency costs by 1 percentage point decreases revenues per employee by almost 11,000 kunas (roughly 1400 euros), which is around 2% of average revenues per employee. This is in line with research by Cadot [58], who estimated the agency costs of vertical integration in the French wine industry and found that agency costs of vertical integration are close to 2.5% of a firm's revenues. Moreover, from the first step (Equation (1)), we could conclude that the estimated coefficient for reputation is the only one that is significant and positively affects revenue per employee. This is consistent with the research findings of Benfratello, Piacenza, and Sacchetto [42], although their dependent variable was wine price. When we controlled for region, we could plainly see that the revenues per employee are much higher for producers in the Adriatic region and that agency costs are lower since, when controlling for regional effect, we could see that the agency costs coefficient is lower. These results were corroborated by descriptive statistics, where we found that the difference between agency costs for producers in the Adriatic and the Continental region is 6%.

The findings of this research are not conclusive. However, they point to some policy considerations that may deserve further research. Empowering small agricultural producers through sectorial programs might prove to be a faster and more efficient path for promoting sustainable development. The support of direct sales can be achieved not only through tax policy but also and more likely by strengthening the financial and educational base of wine producers. Measures to increase reputation and market visibility may be delivered through support to other industry sectors, such as tourism (destination tourism, thematic wine-tours with cellars as points of sale, the possible regional branding of wine

varieties). We still need to build a more detailed insight into consequences of eventual industry consolidation.

The advice that can be extended to wine-producers should focus on building reputation. The scope of our study did not allow for estimations of what the proper level of investment would be, what factors affect payback levels, and how long it takes to see any returns on investment. These aspects may be left for future research. At this moment, it appears that having access to in-house economic support is sound advice despite the fact that an in-house economist (a specialist in understanding markets) might be a rather indirect indicator. Still, calculations have shown its influence on lowering agency costs to be stronger than the influence of the (personal) social capital of the owner, which (theoretically, since the influence is rather low and of lower statistical probability) might count as a somewhat similar factor because both specialist advice and social capital should reflect a wine-producer's capacity to negotiate or make the distributor more inclined to support the interests of a specific wine-producer. Finally, the last important factor seems to lie in developing alternatives. Those producers that naturally (by being based in the touristically active Adriatic region) enjoyed easier access to direct sales were found to present a lower level of dependency and therefore experience better performance. Again, the data employed in the calculations does not distinguish wine producers with a lower dependence and an active strategy of building alternative selling channels from those who are more dependable. However, some respondents in our interviews did indicate that they started (or have already succeeded in) building their own distribution channels in the form of selling points in positions other than their cellars. This is yet another point worth exploring.

## 6. Conclusions

Starting from the premise that market access of Croatian wine producers is commonly provided by intermediaries, we examined several determinants of the bargaining positions of small manufacturing producers that, according to theory and some practical evidence, should affect the nature of the producer-distributor relationship. We modelled the impact of factors that are believed to influence the nature of the relationship between wine producers (as the principal) and distributors (as agents), and we established the mediated impact of agency costs on the financial performance of the wine producer.

To test our hypotheses, we developed a conceptual model in which agency costs act as a mediator variable between the wine producers' input variables and their business performance (output variables). Our analysis was based on a primary dataset gathered via a questionnaire answered by 124 wine producers. Our results clearly indicated that agency costs act as a mediator variable. Considering the idiosyncrasies of the wine sector in general (as well as on a national Croatian level) and based on the results of our analysis, it is clear that wine producers should try to change their sales approach to consumers, that is, they should apply direct sale to insure a better bargaining position towards distributors. Our analysis suggests that a wider range of distribution channels is becoming a prerequisite for mitigating agency-related problems and therefore business development. This could increase profitability in the wine sector, which would, in turn, have long-term benefits for the development of rural areas and the preservation of populations. Moreover, our research highlights the importance and impact of reputation on the size of agency costs in the wine sector, as well as the importance of human capital, that is, the existence of a specialist economist in the wine producing firm. The COVID-19 crisis, which led to a drastically reduced tourist movement and thus a reduction in wine consumption, additionally highlighted the already existing dependence of wine producers on larger merchant distributors. Therefore, the connection between tourism-driven demand for wine should be particularly controlled for to obtain better insight into factors that affect the bargaining position of wine producers.

Several limitations became apparent as we proceeded trough this research:

1. We lacked observations that could explain the position of the distributors; apart from some interviews (not included in this report), we found no insight into factors affecting their economic performance (the value/costs of services they provide, profit margins, how market size and consumer preferences shifts affect their economic standing, etc.). It would be helpful to access contracts and see whether some wine-producers have been able to bargain for particular contract clauses that do not apply to others.

2. We could make no estimations of the average margins and differences in margins earned in direct sales compared to margins earned in selling through market intermediaries. Moreover, we found no insight into data that would allow us to break down agency costs by components (such as monitoring, bonding, residual). Instead, we adapted our research model by reverting to the contributions of authors who have studied vertical agreements in the agricultural value chain [4,5,7,8,34].

Apart from expanding on the topics mentioned in the Discussion section, we recommend for future research to include financial data in combination with, or instead of, self-reported data. Interesting findings could also result from running comparative analyses, with a d similar research design, while employing data from other EU Member States, given that all EU Member States are part of the Common Agricultural Policy.

**Author Contributions:** Conceptualization, J.K., M.K.M. and V.Z.; methodology, J.K., M.K.M. and V.Z.; software, V.Z.; validation, J.K., M.K.M. and V.Z.; formal analysis, J.K., M.K.M. and V.Z.; investigation, J.K. and M.K.M.; resources, J.K. and M.K.M.; data curation, J.K. and V.Z.; writing—original draft preparation, J.K., M.K.M. and V.Z.; writing—review and editing, J.K., M.K.M. and V.Z.; visualization, J.K. and M.K.M.; supervision, M.K.M. and V.Z.; project administration, J.K.; funding acquisition, J.K. All authors have read and agreed to the published version of the manuscript.

**Funding:** This work was supported by the University of Rijeka (UNIRI) Project 20-39.

**Institutional Review Board Statement:** Not applicable.

**Informed Consent Statement:** Not applicable.

**Data Availability Statement:** The data generated or analyzed during this study are available from the corresponding author on reasonable request.

**Conflicts of Interest:** The authors declare no conflict of interest.

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
