# Peer review of "Implications of Mediated Market Access—Exploring the Nature of Vertical Relationships within the Croatian Wine Industry"

_sustainability, doi:10.3390/su14020645_

Round 1
Reviewer 1 Report
This paper deals with an interesting topic about the relationship between wine producers and their distributors. the structure is correct and the findings are pretty interesting. However, several questions must be improved:
- Contribution. There isn't a clear contribution of the paper in the section introduction. Authors should address a propper explanation of "what´s new" in the study., not only explain it with "The aim of this paper is to identify some of the factors...".
- Hypotheses. The formulation of the hypotheses is incorrect. Authors must use different explanations (paragraphs) for each hypothesis.
- Model in figure 1. It is unclear.
- Materials and methods. A deeper explanation of this point is needed, including information about data gathering.
- Discussion and conclusion. This section is not enough to explain the conclusions of the paper. Authors must link their findings with previous research. In addition, future investigations derived from the study are also recommendable to introduce in this section.
Author Response
Dear Reviewer,
Thank you very much for your engagement in reviewing our paper and for indicating what should be corrected to make it better.
Regarding you comments:
- we elaborated more on our contribution to the existining field of knowledge in introduction part (appart from rows 61-62, we added rows 119-131)
- we clearly stated and explained hypotheses in new "Variables and hypotheses" part of the paper
- we redid the Figure 1 and explained it better
- we elabored more the data gathering procedure in Materials and Methods section
- we connected our results with results obtained in papers in the same research field in Discussion part, and added comments about possible future paths to tackle issues that we covered in this paper
Reviewer 2 Report
This review report aims to give some pointers to the paper's authors entitled “implications of mediated market access-exploring the nature of vertical relationships within the Croatian wine industry.” However, unfortunately, I have no choice but to point out that the weaknesses of the paper outrun the strengths. To be specific, the main shortcomings are as follows:
- The abstract is poorly structured in that the vast majority of this section's content is dedicated to setting out the purpose and addressing the subject. Nevertheless, there are no conclusions, and the space devoted to the methodology is just one line. Although it seems to present the obtained results, it is hard to know whether the authors are dealing with the theory or the obtained evidence. Please, restructure the abstract and try to write it up more succinctly and clearly.
- Although the general aim of the paper is addressed (page 2, lines 51-57), the authors do not mention the variables associated with the research objective, nor do they set out the specific goals of the paper. Therefore, please, set out the needs of information and specific research objectives of the paper so that the reader can see the variables associated with each objective.
- It does make any sense to talk about the methodology in the introduction (page 2, lines 89-94). So, please, move this content to the methodology section or delete it.
- Although the authors try to pin down the research gap by pointing out the lack of adopting the producer perspective within the agency theory context (page 1, lines 27-35), the authors do not mention any research variable and, hence, the unavoidable task of addressing the research gap is inconclusive. Therefore, please, address the research gap by saying specific variables and the relationship these variables keep.
- The literature review is shallow, devoid of direction and full of digressions (from page 3, line 100 to page 4, line 187). What is more, the authors neglect to put forward hypotheses. Please, put forward hypotheses by supporting them carefully.
- There is a mismatch between the materials and methods section and the contents included in this section (from page 4, line 189 to page 6, line 256). In my view, the conceptual model should be presented in the review of the literature section. Similarly, the hypotheses should be put forward in the review of the literature section so that all the hypotheses are formulated and supported in the text. Please, restructure and move these contents to the review of the literature section so that you put forward and back up the hypotheses.
- The description of the variables is very poor because it is not technical (page 6, lines 258-267). I would instead the authors bring into focus the sampling procedure, the description of the sample, the survey context, the questionnaire structure and the scales. Please, be focused on methodological content.
- The authors should separate the methodology and the analysis of the results to create two different sections (page 6, line 267). Please, separate two distinct sections.
- The presumed analysis of the results section is not focused on contrasting the hypotheses. Please, contrast the hypotheses empirically.
- Descriptive statistical analyses such as frequencies and correlations are not suitable for contrasting the hypotheses (page 6 line 267, page 7 lines 269-278) and, hence, they are no more than sidetracked contents. So, please, delete it and make good use of the regression results.
- It may not be the case, but the authors seem to use regressions to test mediating effects (page 8, lines 314-page 11 line 380). Nonetheless, the authors should use, for example, a path analysis to test the mediating effects. So, please, make it clear.
- It is rather confusing to create a section called “results” (page 8 line 314) when many results are presented previously. Please, pay attention to how you name the sections and the nature of the contents included.
- As far as discussing and concluding are two different contents, there must be two distinct and separate sections. While the discussion section should be dedicated to comparisons between the obtained results and other papers, the conclusion section should be dedicated to providing practical implications, future lines of research and limitations. So, please, separate both sections and develop each of them accordingly.
I hope these comments can help improve the paper and encourage the authors to move forward.
Author Response
Dear Reviewer,
Thank you very much for your engagement in reviewing our paper and for indicating very precisely what should be corrected to make it better.
Let us first explain what we consider to be new in our study:
First, as far as we are aware there has been no attempts to conduct research on how the bargaining position of Croatian wine producers affects their performance. In doing so, we depart from the positions that the relationship of wine producers towards distributors can be studied by employing the concepts of agency theory, which is also a trajectory that is not usually taken by other researchers. So, instead of analyzing factors that impact on the negotiating power as having a direct impact on performance, we assumed/established that by transferring control over strategic (price and non-price) issues to the distributors, thus the role of agency costs as a mediating variable.
Secondly, we designed our own questionnaire that enabled us to gather first-hand information. We could not anticipate whether the data would confirm our hypotheses, but once calculations were run, they indicated that wine-producers that re more dependable on distributors face higher agency costs. made their bargaining position more fragile, which was confirmed by our calculation. A higher share of sales through intermediaries made their agency costs higher as we can expect their bargaining position to be more fragile. A counter effect was observed over the variable Reputation. These findings were confirmed by our calculation.
The findings also pointed to some issues that demand further investigation to get a cleaner cut picture.
Next, we would like to explain what was done concerning the other advice:
We rephrased the research hypotheses and aligned the figure containing the conceptual along the revised hypotheses.
Materials and methods were explained in more detail. Data was gathered though semi-structured interviews over the spring of 2019.
We added an additional section, under the heading discussion, with comments linking our findings to previous research and explanations, including those on the limitations of our research.
In all parts of our paper, we made extensive revisions, many of which follow the list of shortcomings pointed though items 1.-13. The explanations follow bellow:
- The abstract was revised. Concerning your suggestion that we should determine whether we are developing theory of evidence, we believe our results are more in line with presenting evidence that confirms theoretical expectations. The results also serve to explain what factors can be considered indicative of the level of dependency (maybe even quality of service provided to the wine producer by the distributor, but we did not develop this line of thought to enough detail).
- The variables were described in more detail, both concerning previous research and operationalization for the purpose of our research, as is explained under 3.
- The explanation of the methodology has been substantially revised and restructured. After the first part that leans of the previous section (2. Theoretical background), we first provide an overview of literature sources supporting, followed by two sub-sections: 3.1. Variables and hypotheses and 3.2. Materials and method (explaining how data was gathered and variable operationalization).
- We expect we were able to be more precise in explaining the research gap and justifying our approach by rewriting sections 2 and 3.
- We tried to accommodate this advice though rewriting sections 2 and 3.
- The sampling was random, or rather, wine-makers attending fairs were contacted and questioned in person using pre-prepared semi-structured interviews.
- Advice was taken and used to rewrite section 3.
- The methodology and Analysis are separated under different sections (3 and 4).
- we contrasted the hypothesis with our findings
- we deleted corr. table and upgraded descriptive statistics with three key variables that we didn't include in previous des. stat. table. Also, we elaborated more the statistics part and obtained results. We contrasted the obtained results with the findings of paper in same research field (specially that of Cadot (2015))
- we clearly stated that we use regressions to test mediating effects. This approach is technically different but the result (coefficients) is the same as with using path analysis (we can send proof for that). What we did in our 4-step approach can be called "manual" path analysis.
- we clearly disentagled results and discussion sections (as well as text within those sections, obviously)
- see previous comments
We hope that you are satisfied with our comments. Thanks again for your time
Sincerely,
Authors
Reviewer 3 Report
This paper is to explore what a relationship between the agent cost and wine producer. This paper is also taking about the agency theory how to apply this proposed problem. And therefore, 124 wine producers were interviewed for understanding the real situations.
The results have shown the fruitful findings after the modeling and analysis, it revealed that the proposed concept model has examined.
Only one problem in this paper is the Figure 1 has a mistake. It can not view at a normal situation.
Author Response
Dear Reviewer,
thank You for your comments. We corrected Figure 1.
Kind regards,
Authors
Round 2
Reviewer 1 Report
I appreciate the effort made by the authors in improving the manuscript. The actual version sounds better, but it is still necessary to modify several parts of the paper. Specifically:
- Hypothesis. It is necessary a proper explanation of hypotheses 1 and 2 along with the text. What´s the direction of each?
- Style revision. It is necessary a complete revision of the whole manuscript (format, space, comma, etc.)
Author Response
Dear Reviewer,
we redefined the hypothesis in order to comply with the comments of the second reviewer, but we didn't include direction because that would be in collision with the comments of that reviewer (who had several other comments regarding hypothesis). Also, the text was proofread.
Thanks again for taking your time to review our paper,
Authors
Reviewer 2 Report
This is the second review and it is worth noting that the paper entitled “implications of mediated market access-exploring the nature of vertical relationships within the Croatian wine industry” has been improved yet it is not enough. First, it is true up to the point that the abstract has been revised. Second, the authors have moved some methodological contents from the introduction to the methodology section. Third, the authors have moved the conceptual model from the methodology section to the review of the literature section. Fourth, the authors have separated the analysis of the results section and the methodology section. Fifth, the authors have separated the discussion section and the conclusion section. Sixth, the authors have deleted frequencies and correlations and now the analysis of the results section is more focused. Seventh, the authors have found ground to use linear regressions and, in turn, they justify using this statistical technique instead of performing a path analysis. Equally, they have gone back over the headings’ names to eliminate reiterations.
1. Nevertheless, there remain significant shortcomings and inconsistencies. The authors’ answers to my first review points have been more general than specific to each of my concerns, and it makes it more difficult to see how the paper has been improved. Therefore, let me suggest that the authors give a specific answer to each of my comments, please.
In addition to asking for a specific answer, let me indicate below the remaining shortcomings and inconsistencies as follows:
- The authors point out the general research objective, but they fail to set out the specific research objective. I guess these particular research objectives might be as follows:
On the one hand, to measure how the number of products, production quality, distribution, reputation, social capital and specialist economy influence agency cost. On the other hand, to estimate how agency cost determines the firm performance.
- The research gap is not stated clearly in the introduction. Please, say what we need to research further, what lacks to investigate, what we do not know.
- Although the authors say they have hypotheses, they do not explicitly put forward any hypotheses. Please, find support, formulate and put forward the hypotheses in the review of the literature section. For instance, I guess that the hypotheses might be as follows:
H1 The number of products influences agency cost.
H2 The production quality influences agency cost.
H3 The distribution (Percentage of sales through intermediaries) influences agency cost.
H4 The reputation influences agency cost.
H5 The social capital influences agency cost.
H6 The existence of a specialist economist influences agency cost.
H7 The agency cost influences the firm performance.
- I don’t think in-depth interviews are suitable for this research paper's objectives (line 269). Do you mean just interviews? Do you mean context interviews? As far as in-depth interviews are concerned, there must be a profound psychological examination of sentiments and hidden contents. So please, pay careful attention to use the correct term.
- The paper lacks practical implications. Please, provide managerial implications in the conclusion section.
- Although the authors acknowledge limitations and provide future lines of research, they do not put it in the right section. So, please, move the limitations and future lines of the research from the discussion section to the conclusion section.
- The discussion is devoid of insight. So, please make more effort to gain better insight into your results by comparing your obtained evidence to other authors’ results.
I hope these comments help to improve the paper and encourage the authors to move forward
Author Response
Dear Reviewer,
thanks again for taking Your time reviewing our paper. We will first re-reference to Your comments made in the first review round:
- our abstract now contains clearly stated purpose, main theoretical background, methodology and results.
- the variables are now clearly mentioned, along with the aim (page 2, lines 48-55)
- we did as suggested, i.,e. we moved methodolgy paragraph to the methodology section
- we covered this comment in this (second) round of review, see point 2 below
- we took 5th and other comments either first round of review as well as in this round (see points below)
Regarding the comments from this (second) round of review:
- we added specific goals (page 2, lines 50-55)
- we clearly indicated research gap (page 1 & 2, lines 40-47)
- we formulated the hypotheses are suggested (page 5, lines 195-208)
- data gathering process is explained better (pages 7-8, lines 333-347)
- we added practical implications in the conclusion (page 15, lines 560-575)
- we placed limitation in the conclusion section
- we compared obtained results to other authors’ results in the discussion section, for example, see page 14, comparison with the results of Cadot and Benfratello et al. We also compared our results with that of Newton et al. (page 10), that is, in the Results section because we thought that that particular reference is more appropriate to this section. We focused on papers that cover the same topic as ours. We also added more comments into discussion section.
- we also sent text for proofreading
Best regards,
Authors
Round 3
Reviewer 2 Report
This is the paper's third review, and I have no choice but to acknowledge the authors have improved their paper. Specifically, it is worth noting how they set out specific research objectives in the introduction. Similarly, they pin down the research gap. Likewise, they put forward well-formulated and empirically contrastable hypotheses in the review of the literature section. Furthermore, they enrich the discussion with more insightful thoughts. Finally, not only do they provide practical implications but also limitations and future lines of research.
However, a couple of remaining shortcomings are regarded as follows:
- Although the hypotheses are well formulated and empirically contrastable, I am unsure how well they are supported. What is more, the hypotheses are placed all together instead of being separated with specific supporting contents. Please, support each hypothesis with exact theoretical contents. Moreover, review misspellings, please.
- Let me insist that there must be a desirable match between the used research technique and the research objectives. To be more specific, I do not think the authors carried out 124 in-depth interviews (line 344) for two reasons. First, a proper in-depth interview might take more than an hour to perform. Second, an in-depth interview is a research technique suitable for gaining insight into hidden psychological sentiments. Third, an in-depth interview is unstructured by definition and, hence, on no account is it based on questionnaires. Therefore, the authors have performed “interviews” or “context interviews” instead. Please, revise it.
I hope these comments can help improve the paper and encourage the authors to move forward
Author Response
Dear Reviewer,
we really appreciated all Your comments throughout the review process, because we think that the quality of our paper really improved as a result. Regarding Your comments from the third round of the review:
- we separated the text so that particular sub-hypotheses and/or sub-hyptotheses groups are followed by the literature that supports them, see lines 265-347
- we removed in-depth interview mention (it was mentioned only once, by mistake), definetly we did context interview as we mentioned.
Thanks again and best regards,
The Authors